# Leucine Contributes to Copper Stress Tolerance in Peach (*Prunus persica*) Seedlings by Enhancing Photosynthesis and the Antioxidant Defense System

**DOI:** 10.3390/antiox11122455

**Published:** 2022-12-13

**Authors:** Maoxiang Sun, Suhong Li, Qingtao Gong, Yuansong Xiao, Futian Peng

**Affiliations:** 1State Key Laboratory of Crop Biology, College of Horticulture Science and Engineering, Shandong Agricultural University, Tai’an 271018, China; 2Shandong Institute of Pomology, Tai’an 271000, China

**Keywords:** antioxidant systems, Cu stress, leucine, nitrogen metabolism, peach

## Abstract

Heavy metal contamination has a severe impact on ecological health and plant growth and is becoming increasingly serious globally. Copper (Cu) is a heavy metal that is essential for the growth and development of plants, including peach (*Prunus persica* L. Batsch); however, an excess is toxic. In plants, amino acids are involved in responses to abiotic and biotic stresses, such as water deficit, extreme temperatures, high salinity, and heavy metal stress. However, the role of leucine in the regulation of heavy metal stress is currently unclear. Therefore, we investigated the effects of exogenous leucine on the growth of peach seedlings under Cu stress. Exogenous leucine improved the leaf ultrastructure and ionic balance and increased the chlorophyll content, the net photosynthetic rate, and the maximum photochemical efficiency. Furthermore, it attenuated Cu-stress-induced oxidative damage via a decrease in reactive oxygen species (ROS) and the regulation of the antioxidant and osmotic systems. These effects, in turn, ameliorated the reductions in cell viability, cellular activity, and biomass under Cu stress. Moreover, exogenous leucine increased the activities of nitrate reductase (NR), glutamine synthetase (GS), and glutamic acid synthetase (GOGAT) and thus improved the nitrogen metabolism efficiency of plants. In conclusion, leucine significantly improved the photosynthetic performance and antioxidant capacity, reduced Cu accumulation, and promoted nitrogen metabolism, which in turn improved the resistance of peach seedlings to Cu stress.

## 1. Introduction

Heavy metal pollution of farmland soil is one of the most severe environmental problems in the world [1]. Globally, there are 5 million sites of soil pollution, covering 500 million ha of land in which the soils are contaminated by different heavy metals or metalloids [2]. Heavy metal pollution in soil has a combined worldwide economic impact that is estimated to be in excess of USD 10 billion per year [3]. Heavy metal pollution not only decreases crop yield and quality but also affects human health via the food chain [4]. However, certain heavy metals, such as Cu at lower concentrations, are critical for plant growth and development. Cu is an essential trace element involved in many physiological processes of plants. It acts as a cofactor of enzymes such as the Cu/Zn-superoxide dismutase (Cu/Zn SOD) enzyme and polyphenol oxidase. It is involved in physiological processes such as photosynthesis, cell wall metabolism, and ethylene perception [5]. Cu is a component of the plastocyanin in plant chloroplasts, which is involved in the photosynthetic electron transport process and is also an activator of certain enzymes during chlorophyll formation [6]. Cu deficiency in plants can hinder normal growth, while excess Cu can be toxic. At higher concentrations, Cu ions (Cu^2+^) combine with oxygen to produce free radicals and other reactive oxygen species (ROS), which induce oxidative stress and lead to cell death. Moreover, the accumulation of Cu in the food chain can lead to poisoning in humans due to excessive consumption [7]. In recent years, Cu has become a major environmental pollutant due to overuse in the manufacturing and agricultural industries [8]. Therefore, in order to reduce plant damage due to Cu stress in soil, it is necessary to improve plant tolerance to Cu stress and reduce Cu ion enrichment under Cu stress.

Under normal conditions, the production and removal of reactive oxygen species (ROS) in plants are in dynamic balance. Oxidative stress occurs when the accumulated concentration of ROS exceeds the threshold of plant defense mechanisms under heavy metal stress [9]. Cu stress can induce cells to produce a large amount of ROS, leading to lipid peroxidation of the cell membrane and resulting in an increase in malondialdehyde (MDA) content, a reduction in the selective permeability of the plasma membrane, the leaching of intracellular materials, and then damage to photosynthetic organelles, thereby affecting the normal operation of physiological metabolic processes such as material exchange and photosynthesis [10]. ROS can change the distribution of ions and initiate the expression of nuclear genes in the process of transmitting and amplifying signals so that plants can tolerate various stresses [11]. Amino acids enhance the adaptive responses in plants to various stresses by directly/indirectly influencing the physiological processes or regulating stress-related gene expression and enzyme activities in plants [12]. In plants, leucine is not only a nutrient matrix for cell metabolism but also a signal molecule that can mediate protein metabolism, lipid decomposition, and other biological reactions. Leucine spray at different stages increased biomass and nitrogen content, promoted nitrogen absorption, and improved grain yield in rice [13].

Peach (*Prunus persica* L. Batsch) is one of the most widely cultivated fruit trees in China [14]. Previous studies have found that excessive copper in peach orchard soil can reduce the dry matter content of peach trees, reduce the photosynthetic rate of peach leaves, and reduce the chlorophyll content of leaves, which are not conducive to fruit yield and quality [15]. Therefore, it is crucial to reduce the Cu content in plant parts and alleviate the impact of Cu stress to improve the yield and quality of peach. Several studies have reported the beneficial effects of amino acids in plants exposed to metal stress [16]. In general, amino acid molecules have three major functions in response to heavy metal stress, namely metal binding, antioxidant defense, and signaling [17]. Due to their ability to bind metals, amino acids and their by-products can be used to respond to metal toxicity. Amino acids facilitate the chelation of heavy metal ions in cells and xylem sap, thereby resulting in the detoxification of heavy metals and an increase in plant resistance to toxic metal ions by activating antioxidant systems [18]. However, the role of leucine in the regulation of heavy metal stress is currently unclear. Therefore, in this study, peach seedlings were used as experimental materials to explore the mechanism of leucine alleviating Cu stress in peach seedlings.

## 2. Materials and Methods

### 2.1. Experimental Design

Experiments were performed in the experiment center of Shandong Agricultural University (Tai’an, China) in April 2020 (117°13′ E and 36°16′ N). First, 60-day-old ‘*lu xing*’ peach plants, *Prunus persica* (L.) Batsch., were planted in pots. Seedlings (10 cm tall) grown from peach seeds with the same growth trend and without diseases or insect pests were selected and planted. The pots were cylindrical with an inner diameter of 20 cm and a height of 30 cm. Approximately 2.5 kg of garden soil was placed in each pot, and each treatment was repeated in 60 pots (replicates). The basic physical and chemical properties of the tested soil were as follows: the pH value was 6.68, the alkaline hydrolyzable nitrogen content was 45.65 mg∙kg^−1^, the organic matter content was 12.57 g∙kg^−1^, the available phosphorus content was 35.33 mg∙kg^−1^, and the available potassium content was 83.72 mg∙kg^−1^. The growing conditions for the peach trees were day and night temperatures of 29 °C/23 °C, a natural photoperiod of around 12.5 h, and a constant relative humidity of 30%. The seedlings were maintained following conventional management practices. In the preliminary Cu stress screening test (Appendix A), we found that Cu-induced damage to the peach seedlings began to appear after their treatment with 5 mmol∙L^−1^ CuCl_2_·2H_2_O for 6 days, and leucine (10 mmol∙L^−1^) significantly alleviated Cu stress for 6 days, which met the test requirements. Therefore, in this experiment, the concentrations of 5 mmol∙L^−1^ for CuCl_2_ ·2H_2_O and 10 mmol∙L^−1^ for leucine were chosen. The seedlings were treated as follows: water (control), 10 mmol∙L^−1^ leucine (Leu, Shanghai Yuan ye Biotechnology Co., Ltd., Shanghai, China), 5 mmol∙L^−1^ CuCl_2_·2H_2_O (Cu), and 5 mmol∙L^−1^ CuCl_2_·2H_2_O + 10 mmol∙L^−1^ leucine (Cu+Leu). In this experiment, CuCl_2_·2H_2_O and leucine were uniformly applied to the soil one time at a reagent dosage of 200 mL per tree. All treatments were applied in soil pots at 9 am to avoid excessively high temperatures. On the sixth day after treatment, image indicators were tested with fresh samples, and for the other indicators samples were frozen in liquid nitrogen and placed in an ultra low temperature refrigerator at −80 °C for further testing. Three biological repetitions were included per treatment.

### 2.2. Determination of Endogenous Leucine Content

On the sixth day after treatment, the leaves, stems, and roots were dried and sifted through a 100-mesh sieve. Approximately 0.5 g of the powdered sample was accurately weighed, placed in a 10 mL centrifuge tube, and mixed with 10 mL of ultrapure water. This mixture was ultrasonicated for 30 min and centrifuged at 10,000 rpm for 2 min. The pH of the supernatant was adjusted from 5.3 to 2.2 with 1 mol/L hydrochloric acid. The resultant sample was used to determine the leucine content with an automatic amino acid analyzer (Biochrom, Cambridge, UK) [19].

### 2.3. Determination of SPAD Value, Photosynthetic Rate, and Maximum Photochemical Efficiency

In order to observe the dynamic changes in the physiological indexes of peach seedlings after the treatments, we chose to measure them on the 2nd, 4th, and 6th days after treatment. Chlorophyll fluorescence was measured using an Imaging-PAM chlorophyll fluorometer (HeinzWalz GmbH, Effeltrich, Germany). Before the determination of Fo, the plants were dark-adapted for 30 min. Then, the leaves were adapted to actinic light (250 mmol/m^2^/s), and the maximum fluorescence (Fm) and steady-state fluorescence (Fs) under actinic light were measured with a saturated pulse (3000 mmol/m^2^/s). The variable fluorescence/fluorescence maximum (Fv/Fm), the quantum efficiency of PSII (ϕPSII), the photochemical quenching coefficient (qP), and the non-photochemical quenching coefficient (NPQ) were automatically calculated by the software.

The fifth and sixth uppermost leaves from the top of the plant were selected and fully unfolded. The SPAD values were measured using a chlorophyll tester (SPAD-502 plus, Tokyo, Japan), and the photosynthetic rate was measured using a ciras-3 portable photosynthetic instrument (PP Systems, Hitchin, UK). On the sixth day after treatment, the maximum photochemical efficiency (Fv/Fm) was measured using a portable pulse adjustment fluorometer (Handy PEA, Hansatech, UK). Two leaves per plant and three plants per treatment were used.

### 2.4. Determination of Biomass

Thirty days after the treatment, the peach seedlings were uprooted, washed, and dried off with an absorbent paper. Then, the fresh weights of the aboveground and belowground parts were measured. The dry weights of the aboveground and belowground parts were measured after drying the samples in an oven at 80 °C for 30 min and at 60 °C to a constant weight.

### 2.5. Analysis of Leaf Ultrastructure

On the sixth day after treatment, mesophyll tissue (2 mm × 2 mm) was cut with a blade (avoiding the vein), immersed in 2.5% glutaraldehyde, vacuum-dried, and stored at 4 °C for 24 h. The samples were dehydrated with alcohol, soaked in resin, and heated. The sections were stained and observed under an electron microscope (JEOLTEM-1230EX, Tokyo, Japan) [20].

### 2.6. Determination of Cu Element Contents

On the sixth day after treatment, the powdered peach seedling sample was digested with a combination of nitric acid and perchloric acid, boiled to a constant volume, and filtered through a filter membrane (0.22 μm). The Cu element content in the final sample was analyzed with an atomic absorption spectrometer [21].

### 2.7. Determination of Evans Blue Assay and Reactive ROS

On the sixth day after treatment, the leaf blades and roots were cleaned with ultrapure water, dried with an absorbent paper, and soaked in an Evans blue dye solution (0.25%) for 24 h. The stained leaf blades were washed with ultrapure water, dried, and boiled in a mixture of anhydrous ethanol and glycerin (9:1) to remove the chlorophyll entirely. Images were captured. Then, the cell viability of the roots was quantified with Evans blue using a standard curve. Fluorescence was measured at 600 nm [22].

The production rate of superoxide radicals (O_2_^•−^) and the content of hydrogen peroxide (H_2_O_2_) were determined as described previously [23]. The production rate of O_2_^•−^ was determined by analyzing the formation of nitrite (NO_2_-) from hydroxylamine in the presence of O_2_^•−^. The leaf sample was homogenized and centrifuged. The supernatant was mixed and incubated. The absorbance of the supernatant was read at a wavelength of 530 nm, and a standard curve with NO_2_- was used to calculate the production rate of O_2_^•−^. The content of H_2_O_2_ was determined by monitoring the formation of the hydrogen peroxide–titanium complex. The sample was homogenized with cold acetone. Then, a titanium reagent (15% Ti(SO_4_)_2_) was added to a final concentration of 4%. To precipitate the peroxide–titanium complex, 0.2 mL of concentrated NH_4_OH was added per 1 mL of the reaction mixture. After centrifugation (5 min at 10,000× *g*), the pellet was washed twice with acetone and solubilized in 2 mL of 2 N H_2_SO_4_. The absorbance of the solution was measured at 410 nm.

### 2.8. Determination of Proline, Malondialdehyde, and Antioxidant Enzyme Activities on the Sixth Day after Treatment

The determination of the free proline content was performed according to Bates et al. [24]. Leaf samples (0.5 g) from each group were homogenized in 3% (*w*/*v*) sulfosalicylic acid, and the homogenate was filtered through filter paper. After the addition of acid ninhydrin and glacial acetic acid, the resulting mixture was heated at 100 °C for 1 h in a water bath. The reaction was then stopped using ice bath. The mixture was extracted with toluene, and the absorbance of the fraction with toluene aspired from the liquid phase was read at 520 nm. The proline concentration was determined using a calibration curve and expressed as μmol proline g^−1^ FW.

Lipid peroxidation can be measured by determining the malondialdehyde (MDA) content. The MDA determination followed the method described by Zhao [25] with minor modifications. Briefly, we took 1 mL of the sample supernatant extracted above and added 2 mL of 0.67% thiobarbituric acid (TBA). We used a sample containing only 1 mL of water as a negative control. Next, we placed the samples in a boiling water bath for 15 min, rapidly cooled them by immersion in cold water, and poured them into 10 mL centrifuge tubes. We then centrifuged all tubes at 4000 rpm for 20 min and determined the absorbance of all samples at 600 nm, 532 nm, and 450 nm using a spectrophotometer. The MDA content was determined using the following formula: MDA (μmol·g ^−1^) = [6.452 (A_532_ − A_600_) − 0.56 × A450] × VT/(V0 × W), where VT = the total volume of the extract; V0 = the assay volume; and W = plant fresh weight.

All enzymatic activities were determined at 25 °C and expressed as U g^−1^ protein. The CAT activity was measured by monitoring the decrease in H_2_O_2_ at 240 nm for 1 min at 25 °C. The 3 mL reaction mixture contained 100 μL of enzyme extract and 2.9 mL of sodium phosphate buffer (50 mM, pH 6.0) containing 10 mM H_2_O_2_. The CAT activity was calculated as the amount of enzyme that caused a reduction in the absorbance at 240 nm of 0.01 per minute [25]. The peroxidase (POD) activity was determined by a colorimetric method [25] in a reaction mixture containing guaiacol as the substrate. The POD activity was determined based on the change in absorbance at 470 nm due to the oxidation of guaiacol to tetraguaiacol. The POD activity was defined as the amount of enzyme that caused an increase in absorbance at 470 nm of 0.001 per minute. The superoxide dismutase (SOD) activity was determined based on the ability to inhibit the photochemical reduction in nitroblue tetrazolium (NBT) [25]. The 3 mL reaction mixture was initiated by illumination for 2 min at 25 °C, and the absorbance of blue formazan was measured with a spectrophotometer at 560 nm. One unit of SOD activity (U) was defined as the amount of enzyme that caused a 50% inhibition of the NBT reduction. The APX activity (ascorbate peroxidase) was determined by measuring the oxidation rate of ascorbate at 290 nm according to Zhao [25]. The decrease in the AsA concentration was followed as a decline in the optical density at 290 nm, and the activity was calculated using the extinction coefficient (2.8 mM^−1^ cm^−1^ at 290 nm) for AsA. One unit of APX was defined as the amount of enzyme that breaks down 1 μmol AsA min^−1^. The guaiacol peroxidase (GPX) activity was measured using a modification of the procedure by Zhao [25]. The reaction mixture in a total volume of 2 mL contained 25 mM (pH 7.0) sodium phosphate buffer, 0.1 mM EDTA, 0.05% guaiacol (2-ethoxyphenol), 1.0 mM H_2_O_2,_ and 100 μL of enzyme extract. The increase in absorbance due to the oxidation of guaiacol was measured at 470 nm (E = 26.6 mM^−1^ cm^−1^). The dehydroascorbate reductase (DHAR) activity was determined according to Zhao [25]. The reaction mixture in a total volume of 2 mL contained 25 mM (pH 7.0) sodium phosphate buffer, 0.1 mM EDTA, 3.5 mM GSH, 0.4 mM dehydroascorbate (DHAR), and 100 μL of enzyme extract. The DHAR activity was measured by the formation of ascorbate at 265 nm (E = 14 mM^−1^ cm^−1^). The enzyme activity was expressed as unit g^−1^ min^−1^ FW.

### 2.9. Total RNA Extraction and Quantitative PCR Analysis on the Sixth Day after Treatment

Genes of the antioxidant system were identified from the NCBI database (https://www.ncbi.nlm.nih.gov/, accessed on 14 November 2022). RNA of leaves was extracted using the RNA Extraction Kit (Kangwei Century Technology Co., Ltd, Beijing, China) and reverse-transcribed to cDNA (Takara). The primers used in this study are shown in Appendix A. Quantitative PCR was performed to determine the expression of related genes. The reactions were performed using an Ultra SYBR Mixture (Kangwei, Beijing, China) Kit. Three biological replicates and three technical repeats were used in all qPCR reactions. The relative gene expression levels were calculated using the 2 ^–ΔΔCT^ method [26].

### 2.10. Determination of TTC Assay and Electrolyte Leakage on the Sixth Day after Treatment

The root viability was determined by the 2,3,5-triphenyltetrazolium chloride (TTC) reduction method and expressed as the amount of TTC reduced by per gram of root [25].

The root electrolyte leakage was determined according to a previously described method [27]. Approximately 0.5 g of a root was taken in a test tube, mixed with 20 mL of distilled water, and the vacuum was evacuated three times for 20 min each time. The initial conductivity (S1) of the mixture was measured with a conductivity meter (DDS-11A, Shanghai Kanglu Instrument Equipment Co., Ltd., Shanghai, China). The mixture was then sealed and incubated in a boiling water bath for 10 min. The conductivity was measured after cooling (S2). The conductivity of the distilled water was also measured (S0). Then, the percentage of electrolyte leakage was calculated as follows: electrolyte leakage (%) = [(S1 − S0)/(S2 − S0)] × 100.

### 2.11. Determination of Nitrogen Content and Nitrogen-Metabolism-Related Enzyme Activities in Leaves

On the sixth day after the treatment, the nitrate–nitrogen (NO_3_-N) content of the leaves was determined by salicylic acid colorimetry [25]. Under an acidic condition, the leaf extract was mixed with salicylic acid to form nitrosalicylic acid, whose absorbance was read at 410 nm.

The ammonium–nitrogen (NH_4_^+^-N) content was determined by Nessler’s spectro-photometric method [28]. The leaf extract was mixed with hypochlorite and phenol under a strong alkaline condition to form a water-soluble blue indophenol whose absorbance was measured at 625 nm.

The nitrate reductase (NR) activity in the leaves was determined by sulfanilamide colorimetry [25]. The leaf extract was mixed with sulfanilamide in hydrochloric acid and N-1-naphtyl-ethylenediamine to form a red azo dye whose absorbance was measured at 520 nm.

We determined the glutamine synthetase (GS) activity using the plant GS enzyme activity assay kit and determined the glutamic acid synthetase (GOGAT) activity using the plant GOGAT enzyme activity assay kit (Jiangsu enzyme label Biotechnology Co., Ltd., Jiangsu, China), following the manufacturer’s instructions.

### 2.12. Data Processing and Analysis

We collected three biological replicates for each treatment. Origin version 9.8 was used to conduct all statistical analyses. Duncan multiple range tests, which are included in SPSS version 20.0, were performed to detect any statistically significant differences in the mean values (IBM SPSS, Chicago, IL, USA). The threshold of statistical significance used for all tests was *p* < 0.05.

## 3. Results

### 3.1. Effects of Exogenous Leucine on Growth, Chlorophyll Fluorescence Imaging, Net Photosynthetic Rate, SPAD Value, and Maximum Photochemical Efficiency of Peach Seedlings under Cu Stress

It can be seen from Figure 1a that exogenous Leu can enhance the tolerance of peach seedlings under copper stress. Chlorophyll fluorescence can reflect the photosynthetic efficiency and the degree of stress of plants. We measured the changes in chlorophyll fluorescence (Figure 1b), the net photosynthetic rate (Figure 1c), the SPAD value (Figure 1d), and the maximum photochemical efficiency (Fv/Fm) (Figure 1e) of peach seedlings under Cu stress after leucine treatment. Figure 1a shows that the photosynthetic efficiency of leaves decreased under Cu stress, while the application of Leu improved the photosynthetic efficiency under Cu stress. The net photosynthetic rate, SPAD value, and maximum photochemical efficiency of peach seedlings decreased with an increase in time under Cu stress. However, Leu could alleviate the damage of photosynthetic system under Cu stress. Cu stress reduced the net photosynthetic rate, SPAD value, and maximum photochemical efficiency by 53.8%, 28.2%, and 14.5% compared with control, respectively, in peach seedlings within 6 days. Exogenous leucine increased the net photosynthetic rate, SPAD value, and maximum photochemical efficiency of seedlings under Cu stress by 83.35%, 28.6%, and 8.5% compared with the Cu treatment, respectively, within 6 days. The difference in the effect of Leu on the photosynthetic system was not significant under normal conditions compared to the control.

### 3.2. Effects of Exogenous Leucine on Leaf Ultrastructure of Peach Seedlings under Cu Stress

The ultrastructural changes in the cells of peach leaves under Cu stress and with exogenous leucine are shown in Figure 2a–c. In the control seedlings, mesophyll cells and organelles were visible, and cell walls were intact and smooth. The chloroplasts appeared full and fusiform, and the inner grana lamellae were stacked closely and arranged in order. Few starch granules and plastid globules were observed. Under Cu stress, compared with the control, the starch granule volume increased significantly; starch grains occupied almost half of the chloroplast space. In addition, the number of plastid globules increased, and the chloroplast structure was deformed. With exogenous leucine, the volume of starch granules and the number of plastid globules decreased, and the chloroplast structure was similar to control, even under Cu stress.

### 3.3. Effects of Exogenous Leucine on Antioxidant Enzyme Gene Expression and Enzyme Activities in Peach Seedlings under Cu Stress

We took the logarithm of the relative expression of antioxidant protective enzyme genes based on log_2_, so that the final value > 0 was positive regulation and otherwise it was negative regulation. We found that the relative gene expression of antioxidant protective enzymes in the control and Leu treatments was negative, indicating that the control and Leu treatments did not activate the antioxidant protective enzyme system. However, the Cu treatment and the Cu+Leu treatment could activate the antioxidant protective enzyme system in order to remove ROS in the plant. The relative expression of antioxidant protective enzymes in the Cu treatment with Leu addition was higher than that in the Cu treatment, indicating that Leu could reduce the production of ROS in the plant under Cu stress (Figure 3a). The activities of CAT, SOD, GPx, APX, POD, and DHAR increased by 67.6%, 148.9%, 50.3%, 152.1%, 187.1%, and 181.4% compared with control, respectively, under the Cu+Leu treatment. However, the activities of CAT, SOD, GPx, APX, POD, and DHAR decreased by 29.5%, 28.9%, 14.7%, 6.8%, 41.3%, and 51.3%, respectively, under Cu stress compared with the Cu+Leu treatment, indicating that the production of ROS could be effectively reduced by Leu (Figure 3b–g and Figure 4e,f).

### 3.4. Effects of Exogenous Leucine on Biomass, Leucine Content, Cell Viability, and ROS Accumulation in Leaves of Peach Seedlings under Cu Stress

The fresh weights of the aboveground and belowground plant parts of seedlings under Cu stress were significantly lower than those of the control seedlings after 20 days. However, the fresh weights of the aboveground and belowground parts of seedlings under Cu stress with exogenous leucine were higher than those without exogenous leucine (Figure 4a). These findings indicate that Cu stress inhibited plant growth and reduced the fresh and dry weights of peach seedlings; exogenous leucine alleviated these effects and helped plants retain growth under Cu stress. The endogenous leucine contents in the leaves, stems, and roots of peach seedlings under Cu stress were significantly increased compared to those in the control seedlings. The exogenous leucine further significantly increased the endogenous leucine contents in the leaves, shoots, and roots (Figure 4b).

Evans blue staining was used to analyze the extent of damage in leaf cells after 6 days under Cu stress. The leaves of peach seedlings under Cu stress were darker, with a large percentage of the area stained. Meanwhile, the leaves of peach seedlings under Cu stress with exogenous leucine were lighter, with a smaller percentage of the area stained compared with that of the leaves under Cu stress alone (Figure 4c). These findings indicate that cell viability decreased under Cu stress and exogenous leucine mitigated Cu-stress-induced cell death (Figure 4d). High levels of ROS induced under heavy metal stress disrupt membrane stability and hinder plant growth. Cu stress significantly increased the production rate of O_2_^•−^ and the content of H_2_O_2_. However, exogenous leucine reduced the Cu-stress-induced O_2_^•−^ production rate and the H_2_O_2_ content (Figure 4e,f).

### 3.5. Effects of Exogenous Leucine on Root Biomass, Root Cell Viability, Root Viability, Proline, Malondialdehyde Contents, and Electrolyte Leakage of Peach Seedlings under Cu Stress

Figure 5a and Appendix A show that Leu had little effect on the roots of peach seedlings under normal conditions; however, the root biomass was significantly reduced under Cu stress, and Leu could effectively reduce the reduction in root biomass of peach seedlings. The plant root system is the organ that is directly exposed to heavy metals in soil. Evans blue staining was used to analyze the viability of root cells under Cu stress. The roots were darker under Cu stress, and the color became lighter with exogenous leucine (Figure 5b). In addition, the Cu+Leu treatment significantly increased root cell viability and root viability relative to the Cu treatment (Figure 5c,d). The contents of proline in the peach seedlings under Cu stress were significantly higher than those in the control seedlings (1.3 times higher, Figure 5e). Exogenous leucine significantly reduced the Cu-stress-induced increase in proline contents in peach seedlings. The results indicate that the leucine treatment significantly alleviated the osmotic stress caused by high concentrations of Cu in peach seedlings. The malondialdehyde (MDA) content of seedlings under Cu stress was significantly higher than that of the control seedlings. Exogenous leucine reduced the stress-induced increase in the MDA content; however, the content was significantly higher than that of the control. The results indicate that leucine reduced the MDA content and that electrolyte leakage improved membrane stability in peach seedlings under Cu stress (Figure 5f,g). As can be seen from Appendix A, Cu ions were mainly enriched in the roots of peach seedlings, and the Cu+Leu treatment could significantly reduce the Cu ion content in the roots compared with the Cu treatment.

### 3.6. Effects of Exogenous Leucine on Nitrogen Metabolism in Peach Seedlings under Cu Stress

Under normal conditions, the activities of NR, GS, and GOGAT treated with control and Leu were at high levels from 0 to 6 days. The activities of NR, GS, and GOGAT decreased significantly with time under Cu stress. Although the enzyme activity of leucine-treated seedlings under Cu stress was significantly lower than that of control seedlings, the enzyme activity decreased slightly after 4 days, which was significantly higher than that of pure Cu stress seedlings (Figure 6a–c). Cu stress decreased the nitrate–nitrogen content and increased the ammonium–nitrogen and free amino acid contents in the peach seedlings (Figure 6d–f). The application of leucine to the seedlings under Cu stress increased the nitrate–nitrogen content; however, it was lower than that in the control group. Meanwhile, with exogenous leucine, the ammonium–nitrogen content was the same as that in the control group, while the total free amino acid content was significantly higher than that in control group.

### 3.7. Correlation Analysis of Leucine Content with Osmotic System and Antioxidant System in Peach Seedlings

Figure 7 exhibits a heatmap correlation matrix among different physiological and molecular traits. For the correlation study, the leucine content of the peach seedlings as well as osmotic regulatory system and antioxidant system activity (SOD, POD, CAT, APX, GPx, and DHAR) were evaluated. As can be seen from Figure 7, the leucine content in peach seedlings was positively correlated with plant antioxidant enzyme activities, and the correlation was significantly different and was negatively correlated with the hydrogen peroxide content and the MDA content. The MDA content was negatively correlated with the proline content, the SPAD value, the cell viability of roots, the cell viability of leaves, the APX activity, and the net photosynthetic rate. The hydrogen peroxide content was negatively correlated with the SOD activity, POD activity, DHAR activity, CAT activity, and GPx activity.

## 4. Discussion

Plants are inevitably challenged by various environmental stresses, in particular salt, heat, intense irradiance, and heavy metal stress. Abiotic stress can reduce crop growth, plant leaf area, and photosynthesis rates [29]. Leaves, the central organs of photosynthesis in plants, respond to changes in the external environment or internal metabolism via changes in the ultrastructure, opening angle, aspect ratio, or photosynthesis [30]. It has been reported that a low concentration of Cu^2+^ (<300 mg·kg^−1^) could increase the chlorophyll content of ‘*Hanfu*’ apple seedlings and keep the leaves in a healthy state. However, when the Cu^2+^ concentration exceeded 300 mg·kg^−1^, the plants were stressed, and the chlorophyll content and photosynthetic efficiency of seedlings decreased [31]. Under Cu stress, the net photosynthetic rate of peach seedlings decreased, probably due to the change in chloroplast composition [32]. Chlorophyll a/chlorophyll b were significantly decreased after the application of 600 mg·kg^−1^ Cu^2+^, indicating that a high Cu concentration seriously inhibited the synthesis process of chlorophyll and thus reduced plant photosynthetic efficiency [33]. The maximum photochemical efficiency is a useful indicator of photosystem function and efficiency. The maximum photochemical efficiency is inversely proportional to stress, and any significant change in this value reflects the influence of stress on plants [34]. In this study, the SPAD values of leaves decreased significantly under Cu stress. However, exogenous leucine significantly increased the chlorophyll fluorescence, net photosynthetic rate, SPAD value, and maximum photochemical efficiency (Figure 1b–e).

The morphology and ultrastructure of chloroplasts also directly affect the photosynthetic performance of plants. Several studies have investigated the effect of Cu on the chlorophyll ultrastructure [35,36]. Lin et al. (2008) found that higher concentrations of Cu ruptured the chloroplasts completely with an expanded thylakoid matrix [37]. Ji et al. (2007) also found expanded chloroplasts, distorted grana lamella, and ruptured membranes in *Potamogeton malaianus* under Cu stress [38]. An appropriate concentration of Cu^2+^ can promote plant photosynthesis, but an excessive concentration of Cu^2+^ will inactivate chlorophyll proteins, change the chloroplast ultrastructure, destroy the structure and function of thylakoids, ultimately inhibit photosynthesis, and seriously affect nutrient accumulation [39]. In this study, under Cu stress the cell ultrastructure changed, the chloroplasts were distorted, and the number and volume of starch granules and plastid globules increased. In addition, the cell walls were ruptured, and starch granules occupied almost half of the chloroplasts. After the application of leucine, the number of starch granules and plastid globules decreased significantly, and the chloroplast morphology returned to normal (Figure 2). The possible reason is that exogenous leucine inhibited pigment oxidative decomposition, increased protochlorophyllide reductase activity, promoted chlorophyll synthesis, increased the SPAD value, and maintained the shape of chloroplasts, which in turn improved the photosynthetic system of leaves.

Plants suffering from adverse heavy metal stress, including Cu stress, iron stress, and arsenic stress, produce excess internal reactive ROS, such as superoxide (O_2_-), hydrogen peroxide (H_2_O_2_), and hydroxyl radicals (·OH-), which negatively elicit oxidative stress on cellular structures and metabolism [40,41,42,43]. The antioxidant system is divided into the enzymatic system and the non-enzymatic antioxidant system. These enzymes include SOD, POD, CAT, APX, GPx, and other antioxidant enzymes as well as non-enzymatic antioxidant substances such as GSH, ascorbic acid (ASA), MTs, and proline [44]. Previous studies found that the activities of POD, SOD, and CAT in grape roots increased first and then decreased under different Cu concentrations (0.5, 1, 1.5, and 2 mmol/L) [45]. In this study, we found that the expression levels of antioxidant genes were significantly upregulated when seedlings were exposed to Cu stress, among which the expression levels of *PpCAT*, *PpSOD,* and *PpGPx* were more upregulated. Under Cu stress, the expression of antioxidant genes and the activity of antioxidant enzymes were increased after adding Leu, indicating that Leu could reduce the peroxidation damage under Cu stress (Figure 3). According to Figure 4e,f, compared with the Cu treatment, Cu+Leu can significantly reduce the content of H_2_O_2_ and the production of O_2_- in plants. Therefore, it is speculated that Leu has no obvious effect on the antioxidation system under normal conditions, but under Cu stress Leu can activate the antioxidant system and reduce the oxidative damage caused by ROS.

The AsA-GSH cycle is an important way for plants to respond to stress, mainly through the joint action of multiple enzymatic reactions involving the reduced antioxidant AsA and GSH as well as APX and DHAR so as to realize the process of H_2_O_2_ removal and the regeneration of ASA and GSH and maintain the REDOX homeostasis of cells, improving the stress resistance of plants under stress conditions [46]. Previous studies have shown that the ASA and GSH contents in wheat roots were significantly higher under the high concentration of Cu^2+^ at 1 mmol/L than in the control treatment [47]. Wu et al. found that Taxus chinensis var. chinensis synthesized a large amount of ASA, which was used to remove a large amount of H_2_O_2_ accumulated in the root system to weaken high Cu toxicity [48]. In addition, plants can regulate the osmotic balance by increasing the contents of proline and other osmotic regulators, thus maintaining normal cell metabolism and improving plant stress resistance under heavy metal stress. Our results showed that the relative expression and enzyme activities of *PpAPX*, P*pGPx,* and *PpDHAR* were enhanced by Leu supplementation under Cu stress, which partially explained the enhanced ROS scavenging capacity and the decreased proline and MDA contents in Leu-treated peach seedlings under Cu stress (Figure 3, Figure 4 and Figure 5). As can be seen from the Figure 7, the leucine content in peach seedlings was positively correlated with plant antioxidant enzyme activities, and the correlation was significantly different and was negatively correlated with the hydrogen peroxide content and the MDA content. These results indicate that exogenous leucine reduced the accumulation of ROS in peach seedlings under Cu stress, thus reducing the oxidative damage.

Excessive Cu^2+^ is preferentially accumulated in plant roots, and root length will decrease with an increase in the Cu^2+^ concentration, which will affect the absorption of water and mineral elements by roots and inhibit the growth of the aboveground parts [49]. Michaud et al. found that the root length of wheat decreased by 10%, 25%, and 50%, respectively, when the Cu ion concentration in the root reached 100, 150, and 250–300 mg/kg, indicating that, in a certain range, the higher the Cu ion concentration, the more obvious the inhibitory effect on root growth [50]. In this study, the Cu content was the highest in the roots and the lowest in the stems (Appendix A). In the seedlings treated with leucine under Cu stress, the Cu contents in the roots, stems, and leaves were significantly lower than those in the seedlings under Cu stress alone. In addition, the Cu content in the roots decreased to a greater extent with exogenous leucine. The decrease may be due to the chelating function of amino acids such as leucine and their derivatives [51]. As a result, Cu ions may not be transported through the roots to the aboveground parts. Thus, exogenous leucine also increased the root viability under Cu stress by reducing the accumulation of Cu in the roots and leaves, which in turn improved the root absorption ability, nutrient transportation, and photosynthetic performance, which finally promoted plant growth. Root damage can directly affect the growth and development of plant aboveground parts, in which biomass is usually used as a physiological index to detect the degree of the metal toxicity of plants. Huang et al. found that the biomass of white pomelo did not change significantly under the Cu concentration of 0.5–300 μmol/L, but when the Cu concentration reached 400 μmol/L, the biomass decreased significantly. Studies have shown that Cu stress inhibits plant growth [52]. In Abutilon theophrasti, Cu inhibited root tip cells [53]. Meanwhile, sublethal levels of Cu resulted in lipid peroxidation, which destroyed the membrane structure and affected the root physiological function of beans [54]. In this study, Cu stress significantly reduced the length, area, and volume of roots, the number of root tips, and the number of bifurcations and fibrous roots (Figure 5a, Appendix A). In addition, the fresh and dry weights of the aboveground and belowground plant parts were significantly reduced (Figure 4a). These changes are consistent with a reduction in root viability under Cu stress (Figure 5b–d).

Nitrogen metabolism is the primary source of protein and amino acids in plants [55]. Plants use a series of enzymes, such as NR, GS, and GOGAT, that are involved in nitrogen metabolism to absorb and efficiently utilize nitrogen [56]. NR is a rate-limiting enzyme in the nitrate assimilation pathway, and its activity is sensitive to H_2_O_2_ [57]. In this study, Cu stress significantly increased the H_2_O_2_ content of plants, which led to a decrease in NR enzyme activity. GS and GOGAT are key enzymes that convert inorganic nitrogen to organic nitrogen. The higher content of ammonium–nitrogen in peach seedlings under Cu stress may be related to the decrease in GS/GOGAT activities. The inhibition of the GS/GOGAT pathway hindered the assimilation of inorganic nitrogen into soluble protein [58]. However, a high ammonium–nitrogen content was harmful to peach growth. Cu stress also induced an increase in free amino acids, which may be related to autophagy and accelerated protein degradation [59]. In seedlings with exogenous leucine under Cu stress, the free leucine content increased significantly, probably from exogenous leucine. To maintain an amino acid balance, plants absorb and synthesize more of other amino acids, which further increases the total amount of free amino acids in the peach seedlings. Thus, in this study, leucine retained nitrogen metabolism efficiency in peach seedlings under Cu stress (Figure 6).

## 5. Conclusions

Exogenous leucine improved the leaf ultrastructure, ionic balance, and photosynthetic parameters. Furthermore, it improved the nitrogen metabolism efficiency of plants and attenuated Cu-stress-induced oxidative damage via a decrease in reactive oxygen species (ROS) and the regulation of the antioxidant and osmotic systems. These effects in turn ameliorated cell viability and biomass accumulation with improved resistance of peach seedlings to Cu stress. This study illustrates that leucine alleviates the damage caused by Cu stress in peach seedlings, which may provide more reference data for environmental risk assessments of Cu and make it possible to reuse soils with excessive copper contents caused by heavy metals.

## Figures and Tables

**Figure 1 antioxidants-11-02455-f001:**
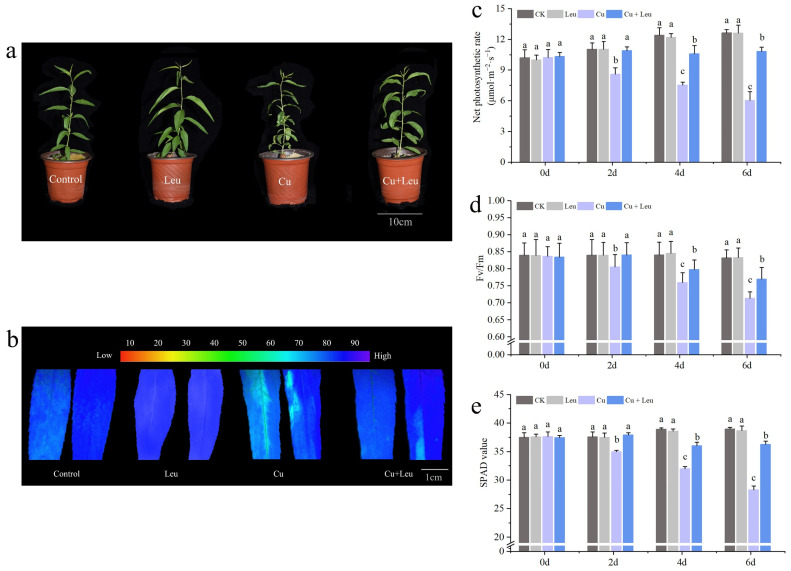
Effects of different treatments on growth, chlorophyll fluorescence imaging, net photosynthetic rate, SPAD value, and maximum photochemical efficiency of peach seedlings. (**a**) Peach seedlings under different treatments; (**b**) chlorophyll fluorescence imaging; (**c**) net photosynthetic rate; (**d**) SPAD value; (**e**) maximum photochemical efficiency. Each data point represents the mean (±SD) of three replicates. Error bars represent standard deviations of the means (*n* = 3). Different lowercase letters indicate significant differences among different treatments (Duncan test, *p* < 0.05).

**Figure 2 antioxidants-11-02455-f002:**
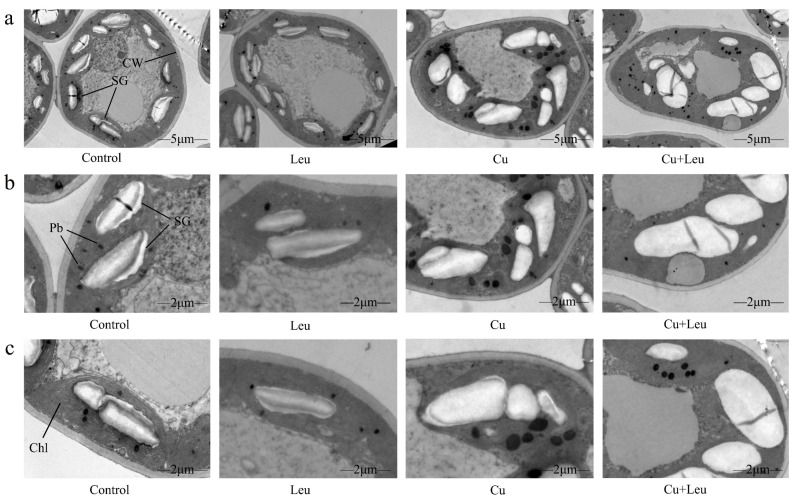
Electron micrographs of leaf mesophyll cells of peach seedlings under different treatments. (**a**) Overall structure of cells with different treatments (bars correspond to 5 μm); (**b**,**c**) Organelle architecture with different treatments (bars correspond to 2 μm). CW: Cell wall; Chl: Chloroplast; SG: Starch granule; Pb: Plastid globule.

**Figure 3 antioxidants-11-02455-f003:**
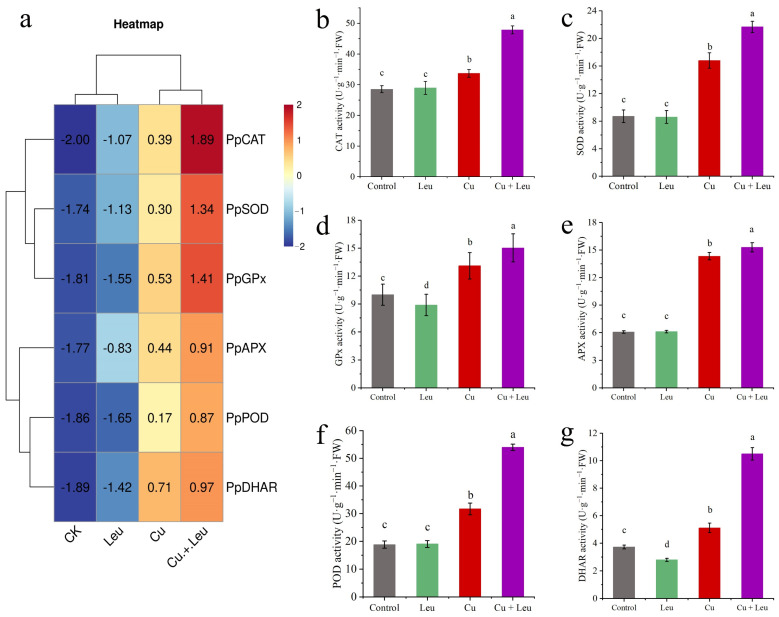
Effects of different treatments on antioxidant protective enzyme gene expression and enzyme activities in peach seedlings. (**a**) The expression of antioxidant enzyme genes; (**b**) CAT activity; (**c**) SOD activity; (**d**) GPx activity; (**e**) APX activity; (**f**) POD activity, (**g**) DHAR activity). Error bars represent standard deviations of the means (*n* = 3). Different lowercase letters indicate significant differences among different treatments (Duncan test, *p* < 0.05).

**Figure 4 antioxidants-11-02455-f004:**
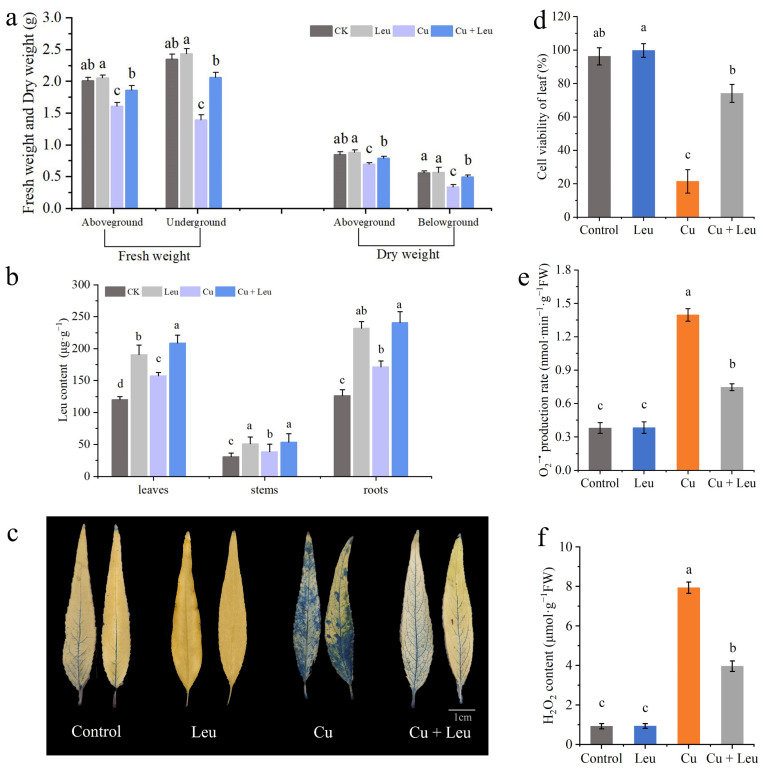
Effects of different treatments on biomass, leucine content, cell viability, and ROS accumulation in leaves of peach seedlings. (**a**) Fresh weight and dry weight of peach seedlings thirty days after the treatment; (**b**) leucine content; (**c**) Evans blue staining of leaves; (**d**) cell viability of leaves; (**e**) leaf O_2_^•−^ production rate; (**f**) H_2_O_2_ content. Error bars represent standard deviations of the means (*n* = 3). Different lowercase letters indicate significant differences among different treatments (Duncan test, *p* < 0.05).

**Figure 5 antioxidants-11-02455-f005:**
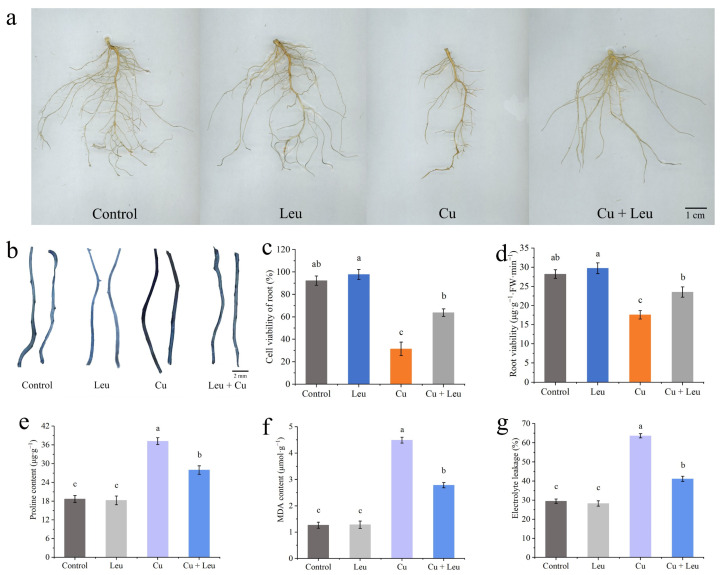
Effects of different treatments on root biomass, root cell viability, root viability, proline, malondialdehyde content, and electrolyte leakage of peach seedlings. (**a**) Root biomass; (**b**) Evans blue staining of roots; (**c**) root cell viability; (**d**) root viability; (**e**) proline content; (**f**) MDA content; (**g**) electrolyte leakage. Error bars represent standard deviations of the means (*n* = 3). Different lowercase letters indicate significant differences among different treatments (Duncan test, *p* < 0.05).

**Figure 6 antioxidants-11-02455-f006:**
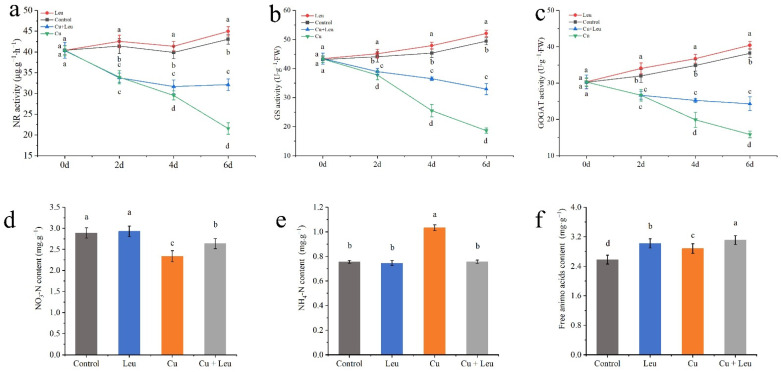
Effects of different treatments on nitrogen metabolism in peach seedlings. (**a**) Nitrate reductase (NR) activity; (**b**) glutamine synthase (GS) activity; (**c**) glutamic acid synthetase (GOGAT) activity; (**d**) nitrate–nitrogen (NO_3_-N) content; (**e**) ammonium–nitrogen (NH_4_+-N) content; (**f**) free amino acid content. Error bars represent standard deviations of the means (*n* = 3). Different lowercase letters indicate significant differences among different treatments (Duncan test, *p* < 0.05).

**Figure 7 antioxidants-11-02455-f007:**
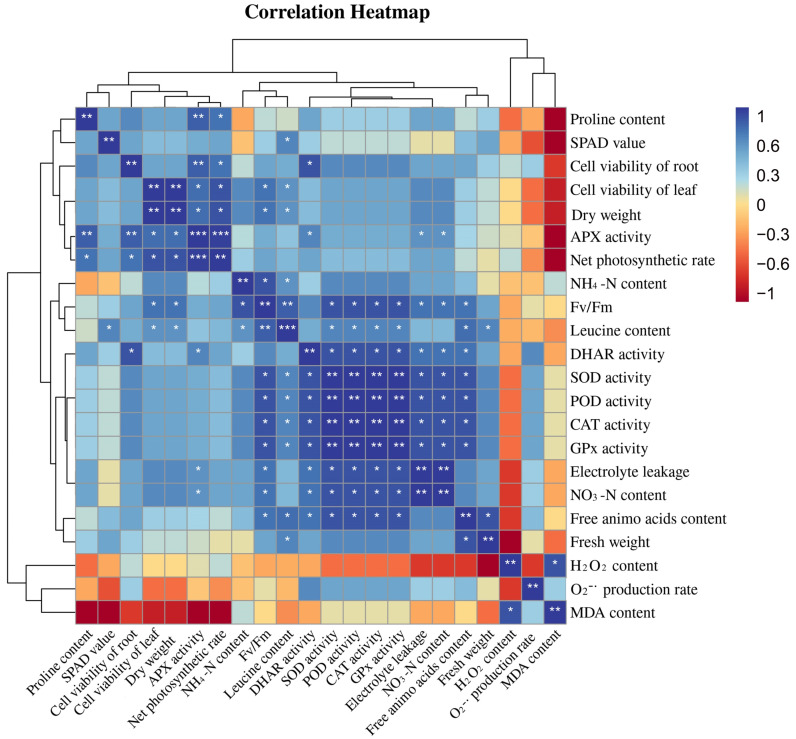
Correlation analysis of leucine content with the osmotic system and the antioxidant system in peach seedlings (Duncan test, * *p* ≤ 0.05, ** *p* ≤ 0.01, *** *p* ≤ 0.001).

## Data Availability

The data are contained within the article.

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
