# Peer review of "Leucine Contributes to Copper Stress Tolerance in Peach (Prunus persica) Seedlings by Enhancing Photosynthesis and the Antioxidant Defense System"

_antioxidants, 2022, doi:10.3390/antiox11122455_

Round 1
Reviewer 1 Report
The Ms ID: antioxidants-2056822 deals with the protective effects of leucine on peach under copper stress. Though the topic is interesting and in the aims of the Journal, there are some flaws in the manuscript and in its organization which should be corrected. Some details of the methods should be specified, but the major flaws are in the Results and Discussion sections. In fact, many sentences of the Results are actually matter of discussion, and some points in the Discussion are not clear. In addition, the section of Conclusion should be totally re-written.
As particular points:
I suggest putting keywords in alphabetical order
L49 change into “Recent”
L54 insert space between oxygen and species
L58-59 as MDA is the result of membrane lipid peroxidation, MDA and lipid peroxidation should be reported as related factors
L59 “leaching” not “extravasation”
L96 how long did the treatments last? It should be specified at this point. In addition, it is not clear to me if some analyses (for example oxidative stress and antioxidant parameters) were performed on fresh material or on stored material. Please, specify
L137 I suggest to use here for Evans Blue assay and below for TTC assay and the term “viability” instead of “activity” as these staining techniques are viability assays
L151 please, add some details to the hydrogen peroxide assay
L163 to get a more precise estimate of the MDA concentration, the non-specific absorbance at 600 nm should be subtracted. If you have done it, you must specify it. Furthermore, the TBA method does not measure only MDA and to be fair it would be better to talk about TBARS (thiobarbituric acid reactive substances)
L164-174 how were the values of U calculated? Basing on extinction coefficient? Please, specify
L189 what do the Authors mean for "extracted three times?
L222 Fig 1a should be omitted and results of these preliminary experiments, not mentioned in the section M&M, should be briefly reported
In legends of Fig.1: as a matter of fact, in the figure the effects of copper and exogenous leucine, also in co-treatment, are reported and this should be mentioned in the legend. Please check also in the other legends
L271-274, 303-306, 321-323, 339-341 these sentences are matter of discussion and they should be moved to that section
L272 I suggest changing “to detoxite” into "to counteract the toxicity of..."
L289 make reference to fig.4
L330 what do the Authors mean for root activity? How was root activity measured? Please add details of method in the section in M&M
L408 this ratio can estimate the investment in antenna and is not a "photosynthetic efficiency index", please correct
L423 write here and throughout the Ms genus and species of plants in italics
L447-448 this sentence is not clear and, as far as I can see, is in contrast with what reported in L453 Please check and explain
L457 GPx is not an enzyme of the AsA-GSH cycle, please correct
L466-467 this seems in contrast with what reported in L447-449 and in Fig.3. Please check and explain
L512 change “transformation” into “assimilation”
L522-525 Conclusions should be re-written as they cannot be a mere list of results
Reviewer 2 Report
The document is attached.

Reviewer 3 Report
The introduction part is very weak and needs special attention. The methodology section needs extensive revision. Moreover, the authors should pay special attention to the discussion which also needs major revision.
The presentation is not well organized in a logical order. For example, in the abstract, lines 12-13 “ Heavy metal contamination has a……..” is a broader aspect. This sentence should appear before the introduction of Copper i.e. line 11-12.
From the abstract through the conclusion, the authors oversimplified the function of ‘Leucine zipper transcription factor’ and amino acid ‘Leucine’, very often using those terms synonymously, thereby misleading the readers through this honest mistake. The authors should go back to the literature and clarify this.
Line 17: avoid the use of the term ‘may’ in this context. If there is strong evidence that leucine acts as a signal, you should first state this in a brief introduction instead of mixing it with the exogenous leucine role that you investigated.
Line 32, 36: In the introduction, from the very beginning the authors tried to establish that heavy metal pollution is a major problem in ‘China’ which underestimate its global effect. Why?
Line 33-34: “ Heavy metals at high levels, including lead, copper, nickel, cadmium, and mercury, affect plant growth and development” should be supported by suitable references including the following: DOI: 10.1111/ppl.13294,
Line 51-53: The statement “Therefore, in order………” is misleading, what is ‘the stress of copper on the soil environment’? How does plant copper tolerance and reduction of copper uptake in plants reduce ‘the stress of copper on the soil environment’? moreover, it is not desirable to reduce the uptake of an essential element (Cu) under optimal conditions, the context, i.e. Cu stress/toxicity should be clearly mentioned. Like this sentence, there are many such instances in the manuscript, the authors should revised them carefully.
Once you introduce the full form of an abbreviation in the first occurrence, it is redundant to further define them or use the full form in the following occurrences. E.g. Copper (Cu), ROS (line 47, 54)
Line 57: ‘will occur’ ?! use present simple tense, not future tense in such context.
Line 72: effect ‘on’, not ‘in’
The authors should provide introduce the severity and effect of copper pollution on peach production/yield in light of literature and statistics, and establish a logical basis for choosing Cu pollution in the context of peach.
Line 74-75: The role of amino acid and leucine in the alleviation of heavy metal stress should be elaborated/ expanded in more detail based on previous literature
Line 75: ‘TF’ should be in full form at the first occurrence followed by an abbreviation in the parenthesis
Line 37-38: mention some more essential metal, such as ‘iron’ and support the statement with suitable references including the following: doi: 10.1016/j.scienta.2020.109205
Line 75, 78: Do not mix ‘leucine zipper transcription factor and exogenous leucine. You should also explain how they are related. from your presentation, it appears that exogenous leucine will directly act as leucine zipper TFs (unbelievable).
Line 48-49: the problem of food chain contamination is universal for all toxic metals and metalloids. Replace ‘copper’ with ‘toxic metal(loid)s’ and support the statement with the following reference: Doi: 10.1016/j.envpol.2021.118475
Line 87: mention the geographical location of the experiment center
Line 87: are these seedlings grown from seeds? Or vegetatively propagated? Provide more details in M&M
Line 89: What does ‘basin’ mean?
Line 90: provide properties of the used ‘garden soil’ in the supplementary
In 2.1 you must provide detail of the treatments, and step-by-step protocol. Did you apply leucine onto the leaf or root/soil? Mention it. If you directly added into the soil, how did you make sure/maintain the concentration of leucine and Copper in soil? You specifically mentioned 10 mmol ∙ L-1 leucine and 5 mmol ∙ L-1 CuCl2 · 2H2O (Cu)! If foliar sprayed, How many times leucine was applied, how much volume each time per plant etc. you should provide more detail so that it can be understood clearly. Otherwise, readers will question the reproducibility of your study.
Moreover, Figure. 1a shows that Leu, Ala (alanine), Ser (serine) and Val (valine) have different effects on the growth and development (line 222) , meaning that you used different amino acid in a preliminary study, but this was one mentioned in M&M. the authors should provide details of this experiment with multiple amino acids, basis of using different concentrations, and finally the selection of leucine as an amino acid with its concentration for further experiment. Add these results in the supplementary. Only ‘pre-experiments’ (line 96), would not suffice it.
Line 91, Line 96: please justify the consistency between “50 pots (replicates)” and “Three biological repetitions”
Line 97: You mentioned that you determined leucine content, but you presented only ‘free amino acid’ content in Fig. 6f. (Line 355, 375) The authors presented data of amino acid, proline but to understand whether exogenous leucine improved endogenous leucine level, the authors should analyze and include data of tissue-specific leucine content.
Line 97, 127, 132, 137, 175, 184, for all these parameters, the authors should mention when each specific parameter `was analyzed, such as ??? days after the treatment and specify tissue type in M&M. In addition, this should be mentioned in respective figure captions/legends
In Figures 1 a, 4c, 5a, 5b, the authors should add ‘scale bars’
Treatment duration should be mentioned in each figure caption
Although exogenous leucine decreased MDA, ROS and electrolyte leakage under Cu stress, it did not increase antioxidant enzyme activity (fig 3). So how oxidative stress was mitigated has not been discussed logically. This article is being considered for the journal ‘Antioxidants’, however, exogenous leucine alleviated oxidative stress by decreasing antioxidant enzyme activity. The authors should justify this discrepancy.
Please improve the discussion with the help of senior colleagues and experts
Line 521: The conclusion is too brief. In addition to the summary, A Conclusion representing some future prospectus related to the present study should be given,
English should be re-checked for grammar
Round 2
Reviewer 1 Report
The revised version of the Ms antioxidants-2056822 results improved. Only a few points remain to be corrected.
In particular:
L88-89 I suggest changing into: “Seedlings (10 cm tall) grown from peach seeds, with the same growth……”
L102-104 I suggest changing into something as ” Therefore, in this experiment, the concentrations of 5 mmol ∙ L-1for CuCl2 ·2H2O and 10 mmol ∙ L-1 for leucine were chosen”
L109-110 I suggest changing into ”…with fresh samples, and for the other indicators samples were frozen in liquid……..”
L146 “On the sixth day after treatment”: as it was clearly specified in the experimental design, here and below it is not necessary to specify it in the different paragraphs and their titles
L205-208 the Authors should write something as: “All enzymatic activities were determined at X °C and expressed as U g- 1 protein. Protein measurement was performed according to ....” In addition, the way the activity calculation has been reported is wrong, since it is necessarily different for different enzymes, as some of them oxidize a substrate, others reduce it. Please check and correct
535-545 I suggest changing into something as “Exogenous leucine improved leaf ultrastructure, ionic balance, and photosynthetic parameters. Furthermore, it improved the nitrogen metabolism efficiency of plants, and attenuated Cu stress-induced oxidative damage via a decrease in reactive oxygen species (ROS) and regulation of the antioxidant and osmotic systems. These effects in turn ameliorated cell viability, and biomass accumulation with improved resistance of peach seedlings to Cu stress.” or similar, avoiding unnecessary repetition of the results.
Reviewer 3 Report
The authors have substantially improved the manuscript with revisions. But there are still language issues. ‘Xylem juice’…what does juice mean? Please use standard technical/botanical terms.
Line 476: “…plants can automatically increase the content of proline …” ‘automatically’? please use standard academic writing/terms. Please pay attention to the English language.
The authors have moved plant phenotype photos with different kinds of amino acid treatment to supplementary materials (replace Figure. 1S with Figure. S1). However, they did not mention what were the concentrations of those 5 amino acids in the figure captions. Please mention it. Moreover, they did not provide the initial screening trial results, where they used different concentrations of leucine and from that, they selected 10 mmol ∙ L-1 leucine in the supplementary. Please add.
The author should add plant phenotype photo with 4 treatments, (Control), 10 mmol ∙ L-1 leucine 104 (Leu, Shanghai Yuan ye Biotechnology Co., Ltd.), 5 mmol ∙ L-1 CuCl2 · 2H2O (Cu), 5 mmol 105 ∙ L-1 CuCl2 · 2H2O + 10 mmol ∙ L-1 leucine (Cu + Leu) in the main body of the manuscript as well, so that readers can get a visual impression about the plant how they look like after Leucine treatment on Cu-treated plants compared with Control, only Leucine, and only Cu treated plants.
The conclusion section has been expanded. But I do not see mention of future perspectives. Such as how the knowledge obtained from this study could potentially be useful, and translation of this knowledge to the field and so on.
